# Beyond MLE: Convex Learning for Text Generation

**Chenze Shao**[*1,2], **Zhengrui Ma**[*1,2], **Min Zhang**[3] **& Yang Feng**[†1,2]
[1] Key Laboratory of Intelligent Information Processing
Institute of Computing Technology, Chinese Academy of Sciences
[2] University of Chinese Academy of Sciences
[3] School of Future Science and Engineering, Soochow University
`chenzeshao@tencent.com, mazhengrui21b@ict.ac.cn`
`zhangminmt@hotmail.com, fengyang@ict.ac.cn`

## Abstract

Maximum likelihood estimation (MLE) is a statistical method used to estimate the parameters of a probability distribution that best explain the observed data. In the context of text generation, MLE is often used to train generative language models, which can then be used to generate new text. However, we argue that MLE is not always necessary and optimal, especially for closed-ended text generation tasks like machine translation. In these tasks, the goal of model is to generate the most appropriate response, which does not necessarily require it to estimate the entire data distribution with MLE. To this end, we propose a novel class of training objectives based on convex functions, which enables text generation models to focus on highly probable outputs without having to estimate the entire data distribution. We investigate the theoretical properties of the optimal predicted distribution when applying convex functions to the loss, demonstrating that convex functions can sharpen the optimal distribution, thereby enabling the model to better capture outputs with high probabilities. Experiments on various text generation tasks and models show the effectiveness of our approach. It enables autoregressive models to bridge the gap between greedy and beam search, and facilitates the learning of non-autoregressive models with a maximum improvement of 9+ BLEU points. Moreover, our approach also exhibits significant impact on large language models (LLMs), substantially enhancing their generative capability on various tasks. Source code is available at `https://github.com/ictnlp/Convex-Learning`.

## 1 Introduction

Text generation is an important field within natural language processing that aims to generate human-like texts for specific tasks. It can be broadly divided into two categories: open-ended and closed-ended text generation. Open-ended tasks encourage the model to produce novel and diverse outputs without a specific expected outcome or structure. Representative tasks in this category include language modeling [41, 7], chatbot [64], storytelling [13], etc. In contrast, closed-ended tasks are more constrained and adhere to specific rules or formats. Representative tasks in this category include machine translation [8, 2], text summarization [45], etc.

In recent years, learning neural probabilistic models with maximum likelihood estimation has become the dominant approach for both open-ended and closed-ended text generation [5, 2, 7]. Maximum likelihood estimation (MLE) is a statistical method used to estimate the parameters of a probability distribution that maximize the likelihood of the observed data [33]. Since directly maximizing the

---

[*]Equal contribution. Order determined by coin flip.

[†]Corresponding author: Yang Feng

37th Conference on Neural Information Processing Systems (NeurIPS 2023).

likelihood can be numerically unstable, it is common to minimize the negative log-likelihood loss function, which is also referred to as cross-entropy loss. It is equivalent to minimizing Kullback-Leibler (KL) divergence [25, 1] between the true distribution and the predicted distribution, which ensures that the optimal predicted distribution is the true data distribution.

While MLE has gained widespread adoption, it does not always align with the objective of text generation, especially for closed-ended text generation tasks such as translation and summarization. In these tasks, the goal of the model is to generate the most appropriate response, rather than producing diverse outputs. For example, in the task of machine translation, though there may exist multiple translations for the same input sentence, we usually want the most accurate and commonly used translation result. Generally speaking, the desired output can be mathematically defined as the output with the maximum probability in the true data distribution, which does not necessarily require the model to estimate the entire data distribution with MLE.

In terms of generating the most probable output, MLE is also suboptimal for current neural text generation models. For autoregressive models, even if the model can perfectly fit the data distribution, it still requires decoding algorithms like greedy or beam search to generate the output, which do not guarantee the exact result with the maximum probability. To our knowledge, only Stahlberg and Byrne [55] proposed an exact decoding algorithm for autoregressive models, but it is too slow for practical applications. The limitation in exact decoding can be overcome by non-autoregressive models [17, 14], which independently predict the output at each position. However, fitting the data distribution by MLE is theoretically beyond the ability of non-autoregressive models [21]. In light of these issues, alternative training objectives should be considered to better address the specific requirements of text generation without incurring the shortcomings associated with MLE.

Based on the analysis above, MLE is suboptimal that it trains the model to estimate the data distribution, which complicates the training and decoding of text generation models. It would be advantageous if the model could converge to a sharper optimal distribution under an alternative loss function, as this would enable autoregressive models to easily find high probability outputs and also allow non-autoregressive models to converge to a better distribution. Exploring loss functions with this characteristic could lead to improved performance and efficiency of neural text generation models, particularly for closed-ended tasks.

In this paper, we propose a novel class of training objectives based on convex functions, which help text generation models capture highly likely outputs without estimating the entire data distribution. Intuitively, the concave shape of log-probability discourages the model from assigning a large prediction probability to a single sample, as the marginal benefit diminishes with increasing probability. If the learning criterion is convex or less concave, then intuitively the model would converge to a sharper distribution, which is the motivation of this work. We further investigate the theoretical properties of the optimal predicted distribution when applying convex functions to the loss. Our findings demonstrate that convex functions can sharpen the optimal distribution, allowing the model to better capture outputs with high probabilities.

Experiments on various closed-ended text generation tasks and models show the effectiveness of our approach. Specifically, it enables autoregressive models to bridge the gap between greedy and beam search, and facilitates the learning of non-autoregressive models with a maximum improvement of 9+ BLEU points. Moreover, our approach also exhibits significant impact on large language models, substantially enhancing their generative capability on various tasks.

## 2 Preliminaries

### 2.1 Maximum Likelihood Estimation

Maximum likelihood estimation (MLE) is a statistical method used to estimate the parameters of a probability distribution that best explain the observed data. This is achieved by maximizing a likelihood function so that the observed data is most probable. Since directly maximizing the likelihood can be numerically unstable, it is common to minimize the negative log-likelihood loss function, also referred to as cross-entropy loss. Given the data distribution $p_{data}$ and a parametric model with parameters $\theta$, MLE training minimizes:

$$\mathcal{L}_{MLE}(\theta) = -\mathbb{E}_{x \sim p_{data}(x)}[\log p_\theta(x)]. \tag{1}$$

MLE can be viewed as an attempt to minimize KL divergence between the true underlying distribution of the data $p_{data}$ and the estimated distribution $p_\theta$ provided by the model [1]. The following equation reveals the relationship between MLE loss and KL divergence:

$$\mathcal{D}_{KL}(p_{data} \| p_\theta) = \sum_x p_{data}(x) \log \frac{p_{data}(x)}{p_\theta(x)} = \mathcal{L}_{MLE}(\theta) - H_{data}, \tag{2}$$

where $H_{data}$ is the Shannon entropy of the data distribution, which remains constant with respect to the model parameter $\theta$. Therefore, the MLE loss and KL divergence share the same minimizer that the estimated distribution $p_\theta$ equals to the true distribution $p_{data}$. By minimizing the MLE loss, the predicted distribution is encouraged to be as close as possible to the true data distribution. In the context of text generation, this ensures that the model learns to generate text that closely resembles the text in the training data.

The above discussion can be extended to conditional scenarios. In such cases, the log-likelihood loss can be expressed as:

$$\mathcal{L}_{MLE}(\theta) = -\mathbb{E}_{c \sim p_{data}(c)}[\mathbb{E}_{x \sim p_{data}(x|c)}[\log p_\theta(x|c)]], \tag{3}$$

where $c$ represents the input context. This extension allows the MLE framework to accommodate a wide range of text generation tasks such as machine translation, summarization, dialogue system, etc.

## 2.2 Text Generation Models

Based on how the sequence probability is factorized, neural text generation models can be broadly categorized into two types: autoregressive (AR) models and non-autoregressive (NAR) models. Autoregressive models generate text sequentially, predicting one token at a time based on the previously generated tokens. In AR models, the probability of generating a sequence $x = (x_1, x_2, ..., x_T)$ is factorized as:

$$p_\theta(x|c) = \prod_{t=1}^{T} p_\theta(x_t | x_{<t}, c), \tag{4}$$

where $c$ represents the input context. With the autoregressive decomposition, AR models can perfectly fit the data distribution if it satisfies $p_\theta(x_t|x_{<t}, c) = p_{data}(x_t|x_{<t}, c)$ for every $x, c, t$. In inference, AR models can perform deterministic decoding like greedy/beam search to generate a high probability output, or sample from the model distribution to generate diverse outputs.

In contrast to autoregressive models, non-autoregressive models [17, 14] generate text in parallel, predicting all tokens simultaneously without conditioning on previously generated tokens. This approach can significantly speed up the generation process, as it removes the sequential dependency between tokens. In NAR models, the generation probability is factorized as:

$$p_\theta(x|c) = \prod_{t=1}^{T} p_\theta(x_t | c). \tag{5}$$

Unlike AR models, NAR models can efficiently find the most likely output by using argmax decoding at each step. However, MLE is beyond the ability of NAR models since they are theoretically unable to fit the data distribution. Huang et al. [21] showed that KL divergence from $p_\theta$ to $p_{data}$ is bounded by a non-negative constant:

$$\mathcal{D}_{KL}(p_{data} \| p_\theta) \geq \mathcal{C} = -H_{data}(x|c) + \sum_{t=1}^{T} H_{data}(x_t|c), \tag{6}$$

The MLE loss is minimized when NAR models achieve the equality by ignoring sequential dependency and predicting $p_\theta(x_t|c) = p_{data}(x_t|c)$. Therefore, NAR models trained with MLE often suffer from reduced performance, as they lack the ability to model dependencies between tokens.

## 3 Approach

In this section, we will explore alternative loss functions for the learning of text generation models, which overcomes the limitations of MLE. We begin by introducing a general learning framework that allows arbitrary loss functions. Next, we discuss the benefits of applying convex functions to the loss within this framework. Finally, we use convex functions to construct composite loss functions, which can be used in practical text generation scenarios.

## 3.1 General Learning Framework

For simplicity of notation, we omit condition $c$ in the probabilities, with the data distribution represented as $p_{data}(x)$ and the model predicting the distribution $p_\theta(x)$. The derived theoretical results hold in both unconditional and conditional settings.

First, we introduce the general learning framework for text generation, characterized by the following loss function:

$$\mathcal{L}_f(\theta) = -\mathbb{E}_{x \sim p_{data}(x)}[f(p_\theta(x))], \tag{7}$$

where $f$ is an arbitrary function of the prediction probability $p_\theta(x)$. We impose some basic requirements on $f$: (1) The domain of function $f$ should contain the interval $(0, 1]$; (2) $f$ must be differentiable on the interval $(0, 1]$ since we need to compute its gradient; and (3) $f$ should be an increasing function on $(0, 1]$ to encourage the model to generate the current sample. Under this framework, we can explain maximum likelihood estimation as a special case of $f = \log$, which is a differentiable and increasing function within the interval $(0, 1]$. We also establish some reasonable assumptions:

**Assumption 1** (Countability of Sample Space). *The sample space $\mathcal{X}$ is countable, which allows us to enumerate all samples in a systematic way. Note that $|\mathcal{X}|$ can be either finite or infinite.*

**Assumption 2** (Distinctness of Sample Probabilities). *In the data distribution $p_{data}$, the probabilities of all samples are distinct, which allows us to arrange samples in a strictly descending order of sample probabilities.*[3]

Assumption 1 naturally holds in text generation tasks due to the inherent discreteness of textual data. With a countable sample space and probabilities lying in a dense subspace of real number, it is reasonable to assume the distinctness of sample probabilities. While Assumption 2 is not strictly necessary, removing it would introduce many corner cases that would complicate the subsequent analysis. In the following, we will assume that Assumptions 1-2 always hold, and we arrange the samples such that $p_{data}(x_1) > p_{data}(x_2) > \cdots > p_{data}(x_i) > \cdots$. Since the sample space $\mathcal{X}$ is countable, the loss function in Equation 7 can be reformulated as follows:

$$\mathcal{L}_f(\theta) = -\sum_{i=1}^{|\mathcal{X}|} p_{data}(x_i) \cdot f(p_\theta(x_i)). \tag{8}$$

In this framework, our primary focus is to analyze the probability distribution $p_\theta$ that the model is inclined to predict when the loss function is $\mathcal{L}_f$. We use $p_f$ to denote the optimal distribution that minimizes the loss $\mathcal{L}_f$, which represents the expected outcome of the model. If $\mathcal{L}_f$ has multiple optimal distributions, we use $p_f$ to denote an arbitrary optimal distribution. This choice does not harm the generality of our analysis, as the subsequent discussion is applicable to all optimal distributions. Currently, it is only established that the optimal distribution for the MLE loss $\mathcal{L}_{\log}$ is the data distribution $p_{\log} = p_{data}$. For other loss functions, the following theorem reveals a general property of the optimal distribution. With samples organized in descending order of their probabilities in the data distribution, i.e., $p_{data}(x_1) > p_{data}(x_2) > \cdots > p_{data}(x_i) > \cdots$, the optimal distribution of an arbitrary function $f$ maintains this order as $p_f(x_1) \geq p_f(x_2) \geq \cdots \geq p_f(x_i) \geq \cdots$. The proofs for the theorems presented in this paper can be found in Appendix A.

**Theorem 1.** *Given an arbitrary differentiable and increasing function $f$, the optimal distribution $p_f$ satisfies $p_f(x_1) \geq p_f(x_2) \geq \cdots \geq p_f(x_i) \geq \cdots$.*

In the following, we will further explore the properties of optimal distributions associated with specific loss functions.

## 3.2 Loss with Convex Function

In certain text generation scenarios that require precise and deterministic outputs, it is beneficial for the model to converge to an optimal distribution that is sharper than the data distribution. In

---

[3]When $\mathcal{X}$ is countably infinite, an arbitrary sequence of sample probabilities forms a convergent series since their sum is 1. This guarantees the existence of a maximum point in the series, ensuring that the sample probabilities can be arranged in a strictly descending order.

this section, we demonstrate that this objective can be achieved by employing convex functions as learning criterion.

The MLE loss function is based on log-probability, which is a concave function whose gradient decreases as the probability increases. The concave shape of the learning criterion prevents the model from assigning a large prediction probability to a single sample, since the marginal benefit diminishes as the probability increases. If function $f$ is convex, then intuitively the model would converge to a sharper distribution. The following theorem validates this intuition, which shows that the optimal distribution $p_f$ is a one-hot distribution when $f$ is convex.

**Theorem 2.** *If $f$ is an increasing convex function on $[0, 1]$, then the optimal distribution $p_f$ is a one-hot distribution that $p_f(x_1) = 1$ and $p_f(x_i) = 0, i > 1$.*

The one-hot characteristic of the optimal distribution is advantageous for text generation models seeking precise and deterministic outputs. For autoregressive models, they do not need the computationally expensive beam search decoding any more if the model distribution is nearly one-hot. For non-autoregressive models, they suffer from reduced performance under MLE due to their inability to fit the data distribution. However, fitting a one-hot optimal distribution is well within their capabilities, allowing these models to generate high-quality outputs.

However, the direct application of loss with convex functions in training text generation models comes with an inherent limitation, impeding its practical utility. Specifically, the gradient of the parameter $\theta$ tends to be very small when the prediction probability approaches 0, thereby rendering the training process inefficient. The gradient of $\theta$ can be formulated as follows:

$$
\begin{aligned}
\frac{\partial \mathcal{L}_f(\theta)}{\partial \theta} &= -\mathbb{E}_{x \sim p_{data}(x)}[f'(p_\theta(x)) \cdot \frac{\partial p_\theta(x)}{\partial \theta}] \\
&= -\mathbb{E}_{x \sim p_{data}(x)}[f'(p_\theta(x)) \cdot p_\theta(x) \cdot \sum_{t=1}^{T} \frac{\partial \log(p_\theta(x_t))}{\partial \theta}],
\end{aligned}
\tag{9}
$$

where we have omitted the autoregressive history condition of $p_\theta(x_t)$ for simplicity. The equation above indicates that the gradient is proportional to the sentence probability $p_\theta(x)$. In text generation models, the sentence probability $p_\theta(x)$ is the product of token probabilities $p_\theta(x_t)$, which causes $p_\theta(x)$ to be typically close to 0, especially when the model is newly initialized.

To counter this effect, the gradient $f'(p_\theta(x))$ would need to approach infinity as $p_\theta(x)$ approaches 0. For instance, the log-probability function has the gradient $\frac{1}{p_\theta(x)}$, which offsets the impact of $p_\theta(x)$ such that $f'(p_\theta(x)) \cdot p_\theta(x) = 1$. However, for an increasing convex function $f(p_\theta(x))$ whose gradient increases with $p_\theta(x)$, its gradient must be bounded when $p_\theta(x)$ approaches 0, leading to an extremely small gradient update for the parameter $\theta$ during training. This inherent limitation of loss with convex functions poses a significant hurdle to their practical applications.

### 3.3 Loss with Convex-composition Function

#### 3.3.1 Theoretical Analysis

In the preceding discussion, we illustrate that while convex functions can induce a desirable one-hot optimal distribution, their inherent limitations during training pose significant impediments to practical applications. Consequently, we consider a relaxation of the convexity requirement, with the objective of rendering the function $f$ less concave. This approach aims to obtain an optimal distribution that is sharper than $p_{data}$, thereby providing a practical solution that augments model performance without sacrificing training feasibility.

The standard loss function in maximum likelihood estimation is the negative log-probability, where log-probability is a concave function that yields a smooth optimal distribution. To render the learning criterion less concave, we propose a convex-composition approach that combines a convex function $f$ with the original concave function $g$. This composition yields the following loss function:

$$
\mathcal{L}_{fg}(\theta) = -\sum_{i=1}^{|\mathcal{X}|} p_{data}(x_i) \cdot fg(p_\theta(x_i)),
\tag{10}
$$

where $f$ is an increasing convex function and $g$ is an increasing concave function. The objective of this composition is to moderate the concavity of the overall loss function, thereby allowing for a sharper optimal distribution. The subsequent theorem and corollaries outline the theoretical properties associated with the optimal distribution under this function composition framework.

**Theorem 3.** *Let $f$ be an increasing convex function and $g$ be an increasing concave function. Then, there exists a positive integer $m$ such that the following inequalities hold:*

1. *$p_{fg}(x_i) \geq p_g(x_i)$ for all $i < m$,*

2. *$p_{fg}(x_i) \leq p_g(x_i)$ for all $i \geq m$.*

**Corollary 1.** *The Shannon entropy of $p_{fg}$ is less than or equal to the Shannon entropy of $p_g$.*

**Corollary 2.** *For any $n \in \{1, 2, ...\}$, the sum of the probabilities of the $n$ most probable samples increases: $\sum_{i=1}^{n} p_{fg}(x_i) \geq \sum_{i=1}^{n} p_g(x_i)$.*

Theorem 3 indicates that the convex-composition loss function tends to allocate higher probabilities to the more probable samples, while simultaneously diminishing the probabilities assigned to less probable ones, resulting in a sharper optimal distribution. Corollary 1 quantitatively establishes this observation, demonstrating that the incorporation of a convex function into the loss function effectively sharpens the optimal distribution, as evidenced by a reduction in the Shannon entropy of $p_{fg}$ compared with $p_g$. Furthermore, Corollary 2 reveals an increase in the cumulative probability of the $n$ most probable samples. Consequently, text generation models are better equipped to capture the highly probable outputs without explicitly modeling the data distribution.

In the above analysis, we only assume the original loss $g$ to be an increasing concave function. By imposing specific conditions on $g$, we can derive more desirable properties from the optimal distribution $p_{fg}$, as demonstrated in the subsequent theorem:

**Theorem 4.** *Let $f$ be an increasing convex function and $g$ be an increasing concave function. If $g$ satisfies $g'''(x) \cdot g'(x) \geq g''(x)^2 > 0$ for all $x \in (0, 1)$, then the difference between $p_{fg}$ and $p_g$ exhibits a monotonic order: $p_{fg}(x_1) - p_g(x_1) \geq p_{fg}(x_2) - p_g(x_2) \geq ... \geq p_{fg}(x_{m-1}) - p_g(x_{m-1}) \geq 0$, where $m$ is the positive integer described in Theorem 3.*

This theorem provides a more granular description of the relative difference between $p_{fg}$ and $p_g$. When $p_{fg}$ decreases the probabilities assigned to less probable samples, it tends to reallocate this probability mass to the most probable samples. This enables text generation models to more accurately capture the most probable outputs. Note that the condition $g'''(x) \cdot g'(x) \geq g''(x)^2 > 0$ is not overly restrictive. For instance, the loss function $g = \log$ in MLE readily fulfills this condition:

$$\log'''(x) \cdot \log'(x) - log''(x)^2 = \frac{2}{x^3} \cdot \frac{1}{x} - (-\frac{1}{x^2})^2 = \frac{1}{x^4} > 0. \tag{11}$$

### 3.3.2 Practical Applications

The preceding theoretical analysis highlights the effectiveness of function composition. Here we turn to its practical applications and give some examples of convex-composition loss functions. The loss function in maximum likelihood estimation is typically the log-probability, and length normalization is often applied in practical usage, resulting in the loss $g(p_\theta(x)) = \frac{\log(p_\theta(x))}{T}$, where $T$ denotes the sentence length. Common choices for increasing convex functions on $(-\infty, 0]$ include the exponential function $f(x) = e^{kx}, k \geq 0$ and the power function $f(x) = -(-x)^k, 0 \leq k \leq 1$. Through function composition, we can derive the following losses:

$$fg(p_\theta(x)) = \begin{cases} p_\theta(x)^{\frac{k}{T}}, & f(x) = e^{kx} \\ -(-\frac{\log(p_\theta(x))}{T})^k, & f(x) = -(-x)^k \end{cases}. \tag{12}$$

The gradient of the convex-composition function is $f'(g(p_\theta(x))) \cdot g'(p_\theta(x))$. Compared to the gradient of the original loss $g'(p_\theta(x))$, it has an additional term $f'(g(p_\theta(x)))$ that can be interpreted as a weight for the loss. Given that $f$ is a convex function and $g$ is an increasing function, the weight $f'(g(p_\theta(x)))$ is larger for more probable samples, thereby directing the model's focus towards

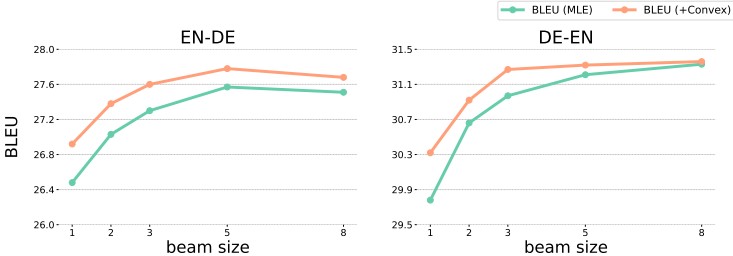

Figure 1: Translation quality (BLEU) of autoregressive model as beam size varies on WMT14 EN↔DE test set.

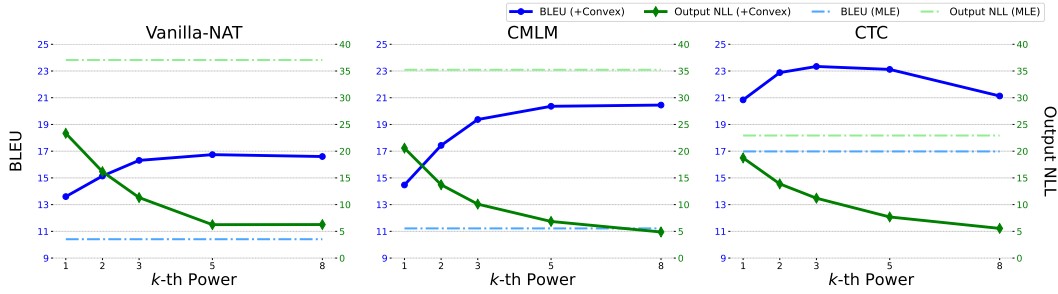

Figure 2: Translation quality (BLEU) and prediction confidence (Output NLL) of different NAT models as the exponent $k$ varies on WMT14 EN-DE test set.

generating outputs with high probabilities. Specifically, the loss weights $f'(g(p_\theta(x)))$ associated with Equation 12 are:

$$f'(g(p_\theta(x))) = \begin{cases} k \cdot p_\theta(x)^{\frac{k}{T}}, & f(x) = e^{kx} \\ k \cdot (-\frac{\log(p_\theta(x))}{T})^{k-1}, & f(x) = -(-x)^k \end{cases} , \tag{13}$$

where the exponential function weights the sample by the prediction probability, and the power function weights the sample by the log-probability.

In practical applications, label smoothing [59, 63] is a widely used regularization technique for text generation models. The smoothing loss and log-probability loss are typically combined using a fixed hyperparameter $\epsilon_{ls}$. To preserve the ratio of smoothing loss to log-probability loss, we also apply the weight $f'(g(p_\theta(x)))$ to the smoothing loss before interpolating it with the convex-composition loss.

## 4 Experiments

To validate the practical advantages of loss functions with sharper optimal distributions, we conduct experiments on basic autoregressive (AR) models, non-autoregressive (NAR) models, and large language models (LLMs). We evaluate their performance on two representative closed-ended text generation tasks, including neural machine translation and text summarization. Following the theoretical analysis in previous sections, we combine the exponential function with standard log-probability, i.e. $\mathcal{L}_f(\theta) = -\mathbb{E}_{x \sim p_{data}(x)}[p_\theta(x)^{\frac{k}{T}}]$, as our training objective in the following experiments. We have also attempted to combine the power function with log-probability as training objective. We found that the power form encountered some difficulties during training, leading to worse performance compared to the exponential form. Due to the space limit, we leave the results under this setting in Appendix E.

Our theoretical analysis suggests that the model trained by convex-composition loss tends to predict a sharper distribution, in which the probability mass is more heavily allocated to the most probable samples. Such property leads the model becoming more confident about its prediction and facilitates the de-facto maximum a posteriori (MAP) decoding framework in closed-ended text generation tasks. In the following, we will discuss and validate the effects of convexity in the context of AR models, NAR models, and LLMs respectively. More details of settings can be found in Appendix B.

Table 1: BLEU scores of autoregressive models on WMT14 EN↔DE test set with different decoding strategies.

| Model | EN-DE | | | DE-EN | | |
| | greedy | beam5 | Δ | greedy | beam5 | Δ |
|---|---|---|---|---|---|---|
| Transformer [62] | 26.48 | 27.57 | 1.09 | 29.78 | 31.21 | 1.43 |
| **Transformer + Convex** | **26.92** | **27.78** | 0.86 | **30.32** | **31.33** | 1.01 |

Table 2: ROUGE scores on CNN/DailyMail and XSum test sets. RG-1, RG-2, RG-L stand for ROUGE-1, ROUGE-2 and ROUGE-L scores.

| Model | CNN/DM | | | XSUM | | |
| | RG-1 | RG-2 | RG-L | RG-1 | RG-2 | RG-L |
|---|---|---|---|---|---|---|
| Transformer [62] | 39.03 | 15.98 | 35.88 | 31.04 | 10.68 | 24.77 |
| **Transformer + Convex** | **39.56** | **16.84** | **36.26** | **31.55** | **11.13** | **25.09** |

## 4.1 Effects of Convexity on Autoregressive Models

In the context of autoregressive models, a model distribution trained with a convex-composition loss tends to exhibit fewer modes and a sharper distribution, thereby facilitating the task of approximate search algorithms in identifying the most likely output. We validate this conjecture by investigating the performance of greedy and beam search when trained with standard MLE and convex-composition loss in translation and summarization tasks. For translation task, we vary the beam size from $\{1, 2, 3, 5, 8\}$, where beam size 1 can be considered as greedy search.

Figure 1 visualizes the results in terms of BLEU [38], with precise numerical values given in Table 1. We observe a consistent improvement in translation quality when using convex-composition losses compared to MLE, and a similar trend is observed in summarization tasks as detailed in Table 2. These results provide experimental support that the composition with convex function promotes those approximate searching algorithms to perform argmax decoding. Meanwhile, Table 1 exhibits a diminishing gap between greedy search and beam search when equipped with convex-composition loss. This outcome can be attributed to the efficacy of the convex function in reducing the complexity of the model distribution, as described in Theorem 3. Such property amplifies the potential of lightweight approximate decoding algorithms within the autoregressive structure, a desirable trait in the context of modern, computation-intensive autoregressive neural networks.

## 4.2 Effects of Convexity on Non-autoregressive Models

Non-autoregressive models face the challenge of multi-modality, where fitting a data distribution with multiple target modes exceeds the capabilities of NAR models. Therefore, the mode collapse property of convex-composition loss would be beneficial to NAR models. Likelihood training will force the model to ignore sequential dependency, resulting in disfluency in its output (e.g., token repetition and omission). In contrast, convex-composition loss would encourage model to allocate most of its probability mass to the best among all proper candidates. Such property is able to help NAR model avoid generating a mixture of modes, thereby alleviating disfluency issues.

To demonstrate its effectiveness, we investigate the performance of convex-composition loss on three representative NAR models, including Vanilla-NAT [17], CMLM [14] and CTC [46]. Considering most of the NAR researches are restricted in the field of translation, we only conduct experiments on translation dataset. In addition to translation quality, we also assess the prediction confidence and generation fluency of NAR outputs. The prediction confidence is measured with negative log-likelihood of its generation and the fluency is measured by an external pre-trained language model [4]. We use the PPL value reported by the language model to quantify the fluency of generation. The exponent hyperparameter $k$ is manipulated to adjust the convexity of our composite loss function.

---

[4]`https://github.com/facebookresearch/fairseq/tree/main/examples/language_model`

Table 3: BLEU and COMET scores on WMT14 EN↔DE test set.

| Model | Speedup | EN-DE | | DE-EN | |
| --- | --- | --- | --- | --- | --- |
| | | BLEU | COMET | BLEU | COMET |
| Transformer [62] | 1.0× | 27.57 | 82.76 | 31.21 | 82.98 |
| Vanilla-NAT [17] | 15.6× | 10.41 | 40.69 | 16.01 | 56.03 |
| **Vanilla-NAT + Convex** | 15.6× | **16.74** | **57.25** | **22.63** | **68.83** |
| CMLM [14] | 15.0× | 11.22 | 43.62 | 15.26 | 56.63 |
| **CMLM + Convex** | 15.0× | **20.45** | **65.99** | **19.11** | **63.54** |
| CTC [46] | 14.7× | 16.98 | 54.77 | 20.53 | 66.26 |
| **CTC + Convex** | 14.7× | **23.34** | **67.38** | **26.68** | **74.75** |

Table 4: Prediction confidence (Output NLL) and generation fluency (External PPL) of CMLM on WMT14 EN-DE test set.

| k-th Power | 1 | 2 | 3 | 5 | 8 |
| --- | --- | --- | --- | --- | --- |
| **Confidence (Output NLL)** ↓ | 20.57 | 13.72 | 10.09 | 6.85 | 4.88 |
| **Fluency (External PPL)** ↓ | 939.34 | 481.08 | 315.54 | 213.84 | 218.68 |

The results are shown in Figure 2 and Table 3. We observe a consistent improvement in translation quality across all NAT models with a maximum improvement of 9+ BLEU points on CMLM. Meanwhile, Figure 2 implies that prediction confidence significantly gains and the gain increases as $k$ gets larger. Such phenomenon reveals a descending trend of model entropy as applying convex function on loss, which is consistent with Corollary 1. More importantly, we note a strong correlation between model entropy and generation fluency in Table 4, providing clear evidence that the mode collapse property of convex function indeed relieves NAR model from multi-modality problem.

### 4.3 Effects of Convexity on Large Language Models

Large language models have demonstrated remarkable capabilities in various applications, including both open-ended and closed-ended text generation tasks. For open-ended tasks, stochastic decoding methods such as temperature sampling are commonly employed to produce responses. In contrast, deterministic decoding methods like beam search are favored for closed-ended tasks like machine translation [22, 69, 30]. Given that the convex-composition loss enhances the model's ability to identify highly probable sentences, incorporating this loss function into the LLMs' training process would be beneficial to closed-ended generation tasks.

To demonstrate its effectiveness, we assess the performance of LLMs in machine translation (Table 5) and summarization (Table 6). Table 5 reveals that the LLaMA-7B model, incorporating convex-composition loss, surpasses the baseline model across all language pairs, achieving an average improvement of 1.84 BLEU. Likewise, the LLaMA-13B model with convex-composition loss outperforms the baseline model in three out of four language pairs. Table 6 further demonstrates the effectiveness of our method in text summarization. Due to memory limitations, we are only able to decode the text summarization dataset using the LLaMA-7B model.

## 5 Related Work

**Alternative Loss Functions** Maximum likelihood estimation has become the dominant approach for learning text generation models, but it also comes with certain limitations. Various alternative loss functions have been proposed to improve the training process from different perspectives. Regarding the exposure bias problem [42] that autoregressive models are exposed to different distributions during training and inference, Bengio et al. [4], Mihaylova and Martins [32], Zhang et al. [72] proposed to reduce this gap by sampling from the model's own predictions during training. Another issue with text generation models is text degeneration: output text may be bland, incoherent, or gets stuck in repetitive loops [20]. To avoid text degeneration, Dieng et al. [9] proposed a learning criterion termed

Table 5: BLEU scores of Alpaca fine-tuned large language models on WMT22 test sets.

| Model | EN-DE | DE-EN | EN-ZH | ZH-EN | AVG |
|---|---|---|---|---|---|
| LLaMA-7B | 25.42 | 17.93 | 13.86 | 13.17 | 17.59 |
| **LLaMA-7B + Convex** | **27.57** | **19.88** | **15.00** | **15.28** | **19.43** |
| LLaMA-13B | **29.35** | 21.74 | 15.58 | 16.27 | 20.74 |
| **LLaMA-13B + Convex** | 28.75 | **22.20** | **16.25** | **20.08** | **21.82** |

Table 6: ROUGE scores of Alpaca fine-tuned large language models on CNN/DailyMail.

| Model | RG-1 | RG-2 | RG-L | AVG |
|---|---|---|---|---|
| LLaMA-7B | 28.66 | 12.49 | 26.37 | 22.51 |
| **LLaMA-7B + Convex** | **32.76** | **14.67** | **30.00** | **25.81** |

reflective likelihood to penalize incoherent outputs, and Welleck et al. [66] proposed unlikelihood training that forces unlikely generations to be assigned lower probability by the model. Additionally, to address the discrepancy between likelihood training and evaluation metrics, loss functions that more directly optimize evaluation metrics are proposed. Ranzato et al. [42] utilized the reinforcement learning technique to train recurrent neural networks with sequence level objectives. Shen et al. [53] proposed to optimize evaluation metrics with minimum risk training. Norouzi et al. [35], Edunov et al. [12] incorporated evaluation metrics into the maximum likelihood training objective. There are also efforts on learning a more focused distribution for text generation models [37, 71, 56]. However, these approaches primarily reformulate the loss function at the word level, which is insufficient for guiding the model towards identifying high-probability sentences at the sentence level. In contrast, our method explicitly trains the model to concentrate on generating highly probable sentences.

**Reinforcement Learning** Our work aligns closely with reinforcement learning (RL) based training techniques for text generation [67, 58]. While RL techniques typically maximize the expected reward by concentrating the probability mass on the sequence with the highest reward, our approach strives to put all the probability mass on the most likely sequence. RL allows for text generation models to optimize discrete evaluation metrics, which have wide usage in text generation tasks, including machine translation [42, 3], text summarization [39], image captioning [44], dialogue generation [27], etc. Furthermore, RL can be integrated with Generative Adversarial Networks [68] and can leverage human feedback for training [57, 36].

**Loss Functions for NAR Models** The limitation of maximum likelihood estimation is amplified in non-autoregressive (NAR) models since they inherently lack the capability to fit the data distribution [21]. To address this issue, researchers have developed loss functions specifically designed for NAR models, guiding them towards generating coherent text. Shao et al. [50, 52], Ding et al. [10] proposed to train NAR models with sequence-level objective functions. Ghazvininejad et al. [15], Du et al. [11] relaxed the alignment restriction in the cross-entropy loss. Shao et al. [51], Shao and Feng [49], Ma et al. [31] proposed n-gram based differentiable training objectives to optimize n-gram prediction accuracy. However, these methods lack theoretical guarantees for the shape of optimal distribution.

## 6 Conclusion

This paper investigates the theoretical properties and practical applications of a novel class of training objectives based on convex functions. Our findings show that convex functions can sharpen the optimal distribution, enabling text generation models to focus on highly probable outputs without having to estimate the entire data distribution. Experiments on various text generation tasks and models verify our theoretical analysis and demonstrate the practical effectiveness of our approach.

## 7 Acknowledgement

We thank the anonymous reviewers for their insightful comments.

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

# A Proofs

## A.1 Proof of Theorem 1

**Theorem 1.** *Given an arbitrary differentiable and increasing function $f$, the optimal distribution $p_f$ satisfies $p_f(x_1) \geq p_f(x_2) \geq \cdots \geq p_f(x_i) \cdots$.*

*Proof.* We prove this theorem by contradiction. Suppose there exist an indice $(i, j)$ such that $i < j$ and $p_f(x_i) < p_f(x_j)$. In this case, we can construct a distribution $p'_f$ with a lower loss than $p_f$, which contradicts the optimality of $p_f$. Specifically, let $p'_f$ be identical to $p_f$ except for the changes $p'_f(x_i) = p_f(x_j)$ and $p'_f(x_j) = p_f(x_i)$. We denote the loss of a model distribution $p$ as $\mathcal{L}_f(p_\theta = p)$ and show that $p'_f$ has a lower loss:

$$\mathcal{L}_f(p_\theta = p'_f) = \mathcal{L}_f(p_\theta = p_f) + (p_{data}(x_j) - p_{data}(x_i)) \cdot (f(p_f(x_j)) - f(p_f(x_i))) < \mathcal{L}_f(p_\theta = p_f). \tag{14}$$

This inequality contradicts the assumption that $p_f$ minimizes $\mathcal{L}_f$, thereby proving the theorem. $\square$

## A.2 Proof of Theorem 2

**Theorem 2.** *If $f$ is an increasing convex function on $[0, 1]$, then the optimal distribution $p_f$ is a one-hot distribution that $p_f(x_1) = 1$ and $p_f(x_i) = 0, i > 1$.*

*Proof.* We prove this theorem by contradiction. Suppose $p_f$ is not the one-hot distribution described above, then there must exist an index $i > 1$ such that $p_f(x_i) > 0$. In this case, we can construct a distribution $p'_f$ with a lower loss than $p_f$, which contradicts the optimality of $p_f$. Specifically, let $p'_f$ be identical to $p_f$ except for the changes $p'_f(x_1) = p_f(x_1) + \alpha$ and $p'_f(x_i) = p_f(x_i) - \alpha$, where $0 \leq \alpha \leq p_f(x_i)$. Then we can calculate the gradient of loss $\mathcal{L}_f(p_\theta = p'_f)$ with respect to $\alpha$:

$$\left. \frac{\partial \mathcal{L}_f(p_\theta = p'_f)}{\partial \alpha} \right|_{\alpha=0} = p_{data}(x_i) \cdot f'(p_f(x_i)) - p_{data}(x_1) \cdot f'(p_f(x_1)). \tag{15}$$

We can analyze the above equation via the following steps:

$$\begin{cases} 1. \ p_f(x_1) \geq p_f(x_i), \text{ from theorem 1} \\ 2. \ f'(p_f(x_1)) \geq f'(p_f(x_i)) > 0, \text{ from step 1 and the convexity of } f \\ 3. \ p_{data}(x_1) > p_{data}(x_i) > 0 \\ 4. \ \left. \frac{\partial \mathcal{L}_f(p_\theta = p'_f)}{\partial \alpha} \right|_{\alpha=0} < 0, \text{ from steps 2,3} \end{cases} \tag{16}$$

The above reasoning shows the loss can be further reduced, which contradicts the assumption that $p_f$ minimizes $\mathcal{L}_f$ and proves the theorem. $\square$

## A.3 Proof of Theorem 3

**Theorem 3.** *Let $f$ be an increasing convex function and $g$ be an increasing concave function. Then, there exists a positive integer $m$ such that the following inequalities hold:*

*1. $p_{fg}(x_i) \geq p_g(x_i)$ for all $i < m$,*

*2. $p_{fg}(x_i) \leq p_g(x_i)$ for all $i \geq m$.*

*Proof.* We prove this theorem by contradiction. Assuming the theorem does not hold, there must exist an indice $(i, j)$ with $i < j$, $p_{fg}(x_i) < p_g(x_i)$, and $p_{fg}(x_j) > p_g(x_j)$. In this case, we can construct a distribution $p'_{fg}$ with a lower loss than $p_{fg}$, which contradicts the optimality of $p_{fg}$.

First, we can establish the following inequality from the optimality of $p_g$:

$$p_{data}(x_i) \cdot g'(p_g(x_i)) \geq p_{data}(x_j) \cdot g'(p_g(x_j)). \tag{17}$$

Assume the above inequality does not hold, then we can further reduce the loss, which contradicts the optimality of $p_g$. Let $p'_g(x_i) = p_g(x_i) - \alpha$, and $p'_g(x_j) = p_g(x_j) + \alpha$. The gradient of loss $\mathcal{L}_{fg}$ with respect to $\alpha = 0$ is $p_{data}(x_i) \cdot g'(p_g(x_i)) - p_{data}(x_j) \cdot g'(p_g(x_j)) < 0$, so we can further reduce the loss with a positive $\alpha$, proving inequality 17 by contradiction.

Then, let $p'_{fg}$ be identical to $p_{fg}$ except for the changes $p'_{fg}(x_i) = p_{fg}(x_i) + \alpha$ and $p'_{fg}(x_j) = p_{fg}(x_j) - \alpha$, where $0 \leq \alpha \leq p_{fg}(x_j)$. We can calculate the gradient of loss $\mathcal{L}_{fg}(p_\theta = p'_{fg})$ with respect to $\alpha$:

$$\left. \frac{\partial \mathcal{L}_{fg}(p_\theta = p'_{fg})}{\partial \alpha} \right|_{\alpha=0} = p_{data}(x_j) \cdot f'g(p_{fg}(x_j)) \cdot g'(p_{fg}(x_j)) - p_{data}(x_i) \cdot f'g(p_{fg}(x_i)) \cdot g'(p_{fg}(x_i)). \tag{18}$$

We can analyze the above equation via the following steps:

$$\begin{cases} 1. \ g'(p_{fg}(x_i)) \geq g'(p_g(x_i)), \text{ from } p_{fg}(x_i) < p_g(x_i) \text{ and the concavity of } g \\ 2. \ g'(p_{fg}(x_j)) \leq g'(p_g(x_j)), \text{ from } p_{fg}(x_j) > p_g(x_j) \text{ and the concavity of } g \\ 3. \ p_{data}(x_i) \cdot g'(p_g(x_i)) \geq p_{data}(x_j) \cdot g'(p_g(x_j)), \text{ from inequality 17} \end{cases} \tag{19}$$

Combining steps 1-3, we obtain:

$$p_{data}(x_i) \cdot g'(p_{fg}(x_i)) \geq p_{data}(x_i) \cdot g'(p_g(x_i)) \geq p_{data}(x_j) \cdot g'(p_g(x_j)) \geq p_{data}(x_j) \cdot g'(p_{fg}(x_j)). \tag{20}$$

Further, the following steps shows that the gradient is less than 0:

$$\begin{cases} 1. \ p_{fg}(x_i) \geq p_{fg}(x_j), \text{ from theorem 1} \\ 2. \ f'g(p_{fg}(x_i)) \geq f'g(p_{fg}(x_j)), \text{ from step 1, the increasing property of } g, \text{ and the convexity of } f \\ 3. \ \left. \frac{\partial \mathcal{L}_{fg}(p_\theta = p'_{fg})}{\partial \alpha} \right|_{\alpha=0} < 0, \text{ from step 2 and inequality 20} \end{cases} \tag{21}$$

The above reasoning shows the loss can be further reduced, which contradicts the assumption that $p_f$ minimizes $\mathcal{L}_f$ and proves the theorem. $\square$

**Corollary 1.** *The Shannon entropy of $p_{fg}$ is less than or equal to the Shannon entropy of $p_g$.*

*Proof.* The Shannon entropy of distribution $p$, denoted by $H_p$, is defined as $H_p = -\sum_x p(x) \log p(x)$. Consider a function $h(\Delta x) = -(x_1 + \Delta x) \log(x_1 + \Delta x) - (x_2 - \Delta x) \log(x_2 - \Delta x)$. It's first-order derivative $h'(\Delta x) = \log(x_2 - \Delta x) - \log(x_1 + \Delta x)$. Assuming $x_1 \geq x_2$, we observe that $h'(\Delta x) < 0$ when $\Delta x > 0$, so the entropy decreases when we reduce $x_2$ and increase $x_1$ accordingly. The transformation from $p_g$ to $p_{fg}$ can be viewed as a series of such adjustments, which implies that the Shannon entropy of $p_{fg}$ is less than or equal to the Shannon entropy of $p_g$. $\square$

**Corollary 2.** *For any $n \in \{1, 2, ...\}$, the sum of the probabilities of the $n$ most probable samples increases: $\sum_{i=1}^n p_{fg}(x_i) \geq \sum_{i=1}^n p_g(x_i)$.*

*Proof.* The theorem guarantees the existence of a positive integer $m$ such that $p_{fg}(x_i) \geq p_g(x_i)$ for all $i < m$ and $p_{fg}(x_i) \leq p_g(x_i)$ for all $i \geq m$. In the case where $n < m$, we have $p_{fg}(x_i) \geq p_g(x_i)$ for all $i$ with $1 \leq i \leq n < m$. This leads to the inequality $\sum_{i=1}^n p_{fg}(x_i) \geq \sum_{i=1}^n p_g(x_i)$. In the case where $n \geq m$, we have $p_{fg}(x_i) \leq p_g(x_i)$ for all $i$ with $m \leq n < i$. Therefore, we can write $\sum_{i=1}^n (p_{fg}(x_i) - p_g(x_i)) = -\sum_{i=n+1}^{|\mathcal{X}|} (p_{fg}(x_i) - p_g(x_i)) \geq 0$. Consequently, the inequality $\sum_{i=1}^n p_{fg}(x_i) \geq \sum_{i=1}^n p_g(x_i)$ also holds. Therefore, in both cases, the corollary is proved. $\square$

### A.4 Proof of Theorem 4

**Theorem 4.** *Let $f$ be an increasing convex function and $g$ be an increasing concave function. If $g$ satisfies $g'''(x) \cdot g'(x) \geq g''(x)^2 > 0$ for all $x \in (0, 1)$, then the difference between $p_{fg}$ and $p_g$ exhibits a monotonic order: $p_{fg}(x_1) - p_g(x_1) \geq p_{fg}(x_2) - p_g(x_2) \geq ... \geq p_{fg}(x_{m-1}) - p_g(x_{m-1}) \geq 0$, where $m$ is the positive integer described in Theorem 3.*

*Proof.* We prove this theorem by contradiction. Assuming the theorem does not hold, there must exist an indice indice $(i, j)$ with $i < j$ and $0 \le p_{fg}(x_i) - p_g(x_i) < p_{fg}(x_j) - p_g(x_j)$. In this case, we can construct a distribution $p'_{fg}$ with a lower loss than $p_{fg}$, which contradicts the optimality of $p_{fg}$. Specifically, let $p'_{fg}$ be identical to $p_{fg}$ except for the changes $p'_{fg}(x_i) = p_{fg}(x_i) + \alpha$ and $p'_{fg}(x_j) = p_{fg}(x_j) - \alpha$, where $0 \le \alpha \le p_{fg}(x_j)$. Then we can calculate the gradient of loss $\mathcal{L}_{fg}(p_\theta = p'_{fg})$ with respect to $\alpha$:

$$\frac{\partial \mathcal{L}_{fg}(p_\theta = p'_{fg})}{\partial \alpha}\bigg|_{\alpha=0} = p_{data}(x_j) \cdot f'g(p_{fg}(x_j)) \cdot g'(p_{fg}(x_j)) - p_{data}(x_i) \cdot f'g(p_{fg}(x_i)) \cdot g'(p_{fg}(x_i)). \tag{22}$$

Our goal is to demonstrate that $\frac{\partial \mathcal{L}_{fg}(p_\theta = p'_{fg})}{\partial \alpha}\bigg|_{\alpha=0} < 0$, which would contradict the assumption that $p_{fg}$ minimizes $\mathcal{L}_{fg}$, thereby proving the theorem.

Given the optimality of $p_g$, we have $p_{data}(x_i) \cdot g'(p_g(x_i)) \ge p_{data}(x_j) \cdot g'(p_g(x_j))$, otherwise we can reduce $p_g(x_i)$ to obtain a lower loss. From Theorem 1, we know that $p_{fg}(x_i) \ge p_{fg}(x_j)$, and because $f$ is convex and $g$ is increasing, we have $f'g(p_{fg}(x_i)) \ge f'g(p_{fg}(x_j))$. Using these inequalities, we obtain a upperbound of equation 22:

$$\begin{aligned}
\frac{\partial \mathcal{L}_{fg}(p_\theta = p'_{fg})}{\partial \alpha}\bigg|_{\alpha=0} &/ (p_{data}(x_i) \cdot g'(p_g(x_i))) \\
&= \frac{p_{data}(x_j) \cdot f'g(p_{fg}(x_j)) \cdot g'(p_{fg}(x_j))}{p_{data}(x_i) \cdot g'(p_g(x_i))} - \frac{p_{data}(x_i) \cdot f'g(p_{fg}(x_i)) \cdot g'(p_{fg}(x_i))}{p_{data}(x_i) \cdot g'(p_g(x_i))} \\
&\le \frac{p_{data}(x_j) \cdot f'g(p_{fg}(x_j)) \cdot g'(p_{fg}(x_j))}{p_{data}(x_j) \cdot g'(p_g(x_j))} - \frac{p_{data}(x_i) \cdot f'g(p_{fg}(x_i)) \cdot g'(p_{fg}(x_i))}{p_{data}(x_i) \cdot g'(p_g(x_i))} \\
&= \frac{f'g(p_{fg}(x_j)) \cdot g'(p_{fg}(x_j))}{g'(p_g(x_j))} - \frac{f'g(p_{fg}(x_i)) \cdot g'(p_{fg}(x_i))}{g'(p_g(x_i))} \\
&\le f'g(p_{fg}(x_j)) \cdot \left( \frac{g'(p_{fg}(x_j))}{g'(p_g(x_j))} - \frac{g'(p_{fg}(x_i))}{g'(p_g(x_i))} \right).
\end{aligned} \tag{23}$$

To demonstrate that $\frac{\partial \mathcal{L}fg(p\theta = p'_{fg})}{\partial \alpha}\bigg|_{\alpha=0} < 0$, we only need to prove the following inequality:

$$\frac{g'(p_{fg}(x_j))}{g'(p_g(x_j))} - \frac{g'(p_{fg}(x_i))}{g'(p_g(x_i))} < 0. \tag{24}$$

Let $\Delta x = p_{fg}(x_i) - p_g(x_i) < p_{fg}(x_j) - p_g(x_j)$. As a result, $p_{fg}(x_j) > \Delta x + p_g(x_j)$, and thus $g'(p_{fg}(x_j)) < g'(p_g(x_j) + \Delta x)$. This allows us to further simplify the inequality:

$$\frac{g'(p_{fg}(x_j))}{g'(p_g(x_j))} - \frac{g'(p_{fg}(x_i))}{g'(p_g(x_i))} < \frac{g'(p_g(x_j) + \Delta x)}{g'(p_g(x_j))} - \frac{g'(p_g(x_i) + \Delta x)}{g'(p_g(x_i))}. \tag{25}$$

To establish that the right-hand side of the above inequality is non-positive, we can apply the logarithm transformation and show the following inequality instead:

$$\log(g'(p_g(x_j) + \Delta x) - \log(g'(p_g(x_j))) \le \log(g'(p_g(x_i) + \Delta x) - \log(g'(p_g(x_i))). \tag{26}$$

Let's denote $h(x) = \log(g'(x))$, $x_1 = p_g(x_i)$, and $x_2 = p_g(x_j)$. The above inequality can be simplified to:

$$h(x_2 + \Delta x) - h(x_2) \le h(x_1 + \Delta x) - h(x_1), \tag{27}$$

where $\Delta x \ge 0$ and $x_2 \le x_1$ according to Theorem 1. The above inequality holds when $h(x)$ is a convex function. The second-order derivative of $h(x) = \log(g'(x))$ is:

$$h''(x) = \frac{g'''(x)g'(x) - g''(x)^2}{g'(x)^2}. \tag{28}$$

Therefore, $h(x)$ is a convex function under the condition $g'''(x) \cdot g'(x) \ge g''(x)^2$. This verifies that $\frac{\partial \mathcal{L}_{fg}(p_\theta = p'_{fg})}{\partial \alpha}\bigg|_{\alpha=0} < 0$, completing the proof by contradiction. $\qquad \square$

# B   Experimental Settings

## B.1   Machine Translation

### B.1.1   Datasets and Metrics

**Datasets** We conduct experiments on widely used translation benchmark: WMT14 English-German (EN-DE, 4.5M), where the validation and test sets are *newstest2013* and *newstest2014* respectively. We apply BPE [48] with 32K merge operations to learn a joint vocabulary on the tokenized data. Considering the major topic of this research is how to learn from a real-world data distribution, we don't apply any tricks that may have an influence on the distribution, e.g., knowledge distillation.

**Metrics** The overall quality of translation is assessed using metrics BLEU [38] and COMET [43].[5] In the case of non-autoregressive models, we additionally quantify the prediction confidence and translation fluency of the generated output. Prediction confidence is measured with negative log-likelihood (NLL) of model generation. A lower NLL value indicates a more focused model distribution and higher prediction confidence. To evaluate translation fluency, we utilize an external pre-trained autoregressive language model. The generated translation is fed to the language model using teacher forcing and the resulting perplexity (PPL) is calculated as a measure of fluency.[6] A lower external PPL score indicates a higher level of fluency.

### B.1.2   Implementation Details

**Architectures** In order to validate the overall efficacy of convex-composition loss, we perform experiments using various model architectures. We adopt Transformer-*base* [63] as our autoregressive baseline and Vanilla-NAT [17], CMLM [14] and CTC [46] as our non-autoregressive baselines. We apply *uniform copy* to construct decoder inputs in Vanilla-NAT and CTC. The decoder length in CTC is set to 2× the source length.

**Training** Although training with convex-composition loss offers the desirable property of optimality, it can encounter gradient vanishing issues during initialization as analyzed previously. To mitigate this, we employ a two-step training approach: MLE pre-training followed by fine-tuning with convex-composition loss. This approach allows us to avoid numerical gradient issues while still benefiting from the optimality achieved through convex composition. For training with convex-composition loss, we set the exponent hyperparameter $k$ to 1 for the autoregressive model and tune it from $\{1,2,3,5,8\}$ on the validation set for non-autoregressive models. Throughout both MLE and convex-composition training, all models are optimized using the Adam optimizer [24] with $\beta = (0.9, 0.98)$ and $\epsilon = 10^{-8}$. Detailed information regarding other training hyperparameters can be found in Table 7.

Table 7: Settings of training hyperparameters on WMT14 EN↔DE dataset.

|  | Transformer | | Vanilla-NAT | | CMLM | | CTC | |
| --- | --- | --- | --- | --- | --- | --- | --- | --- |
|  | MLE | Convex | MLE | Convex | MLE | Convex | MLE | Convex |
| batch size | 32k | 32k | 64k | 256k | 64k | 256k | 64k | 256k |
| learning rate | 7e-4 | 2e-4 | 5e-4 | 3e-4 | 5e-4 | 3e-4 | 5e-4 | 3e-4 |
| warmup steps | 4k | 1k | 10k | 500 | 10k | 500 | 10k | 500 |
| training steps | 200k | 50k | 300k | 10k | 300k | 10k | 300k | 10k |
| dropout | 0.1 | 0.1 | 0.3 | 0.3 | 0.3 | 0.3 | 0.3 | 0.1 |
| weight decay | 0 | 0 | 0.01 | 0.01 | 0.01 | 0.01 | 0.01 | 0.01 |
| label smoothing | 0.1 | 0.1 | 0.1 | 0 | 0.1 | 0 | 0.01 | 0 |
| length loss factor | - | - | 0.1 | 0.01 | 0.1 | 0.01 | - | - |

**Decoding** For the autoregressive model, we set the beam length to 5 by default and tune the length penalty on the validation set unless stated otherwise. For Vanilla-NAT and CTC, we utilize fully non-autoregressive argmax decoding. In the case of CMLM, we employ 5 length candidates and

---

[5]We use checkpoint `Unbabel/wmt22-comet-da` to compute COMET score. It is available at `https://github.com/Unbabel/COMET`.

[6]We use checkpoint `transformer_lm.wmt19.de` to compute the external PPL score. It is available at `https://github.com/facebookresearch/fairseq/tree/main/examples/language_model`.

disable iteration for inference. The decoding speedup is measured with a batch size of 1 on GeForce RTX 3090 GPUs.

## B.2 Abstractive Summarization

### B.2.1 Datasets and Metrics

We conduct experiments on two widely used summarization benchmarks: CNN/DailyMail [18] and Xsum [34]. CNN/DailyMail contains 220K articles from the Daily Mail newspaper and 93K articles from CNN. Each article contains a bullet point summary consisting of multiple sentences. We use the non-anonymized variant following [47, 29]. After the pre-processing, there are 311,971 ⟨article, summary⟩ pairs. XSum consists of 227K online articles from the British Broadcasting Corporation (BBC), containing professionally written single-sentence summaries. After the preprocessing, there are 226,677 ⟨article, summary⟩ data pairs. In order to maintain consistency with previous works [26, 40], we employ GPT-2 tokenizer to tokenize raw CNN/DailyMail data, and Berttokenizer to tokenize raw Xsum data. The summarization quality is measured with ROUGE-1, ROUGE-2 and ROUGE-L [28] as discussed in [6].

### B.2.2 Implementation Details

In our summarization experiments, most of the implementation details of the Transformer align with those used in translation. However, there are a few modifications to ensure consistency with previous work [26]. We apply layer normalization to the embeddings. The attention dropout is set to 0.1, and the weight decay is set to 0.01. We utilize beam search with a size of 4 during decoding. The length penalty, max_len_b, and min_len are set to 2.0, 140, and 55, respectively on CNN/DailyMail dataset. We use a length penalty of 1.2 on Xsum dataset. For CNN/DailyMail dataset, we additionally employ a tri-gram repetition prevention trick.

## B.3 Large Language Models

For the development of LLMs, we utilize LLaMA-7B and LLaMA-13B [61] as our foundation models. We conduct instruction tuning using the Alpaca dataset by GPT4 [65, 60], which comprises 52K instruction-following demonstrations. Instead of the standard cross-entropy loss employed during instruction tuning, we adopt the convex-composition loss of exponential form to fine-tune foundation models.

The generative capability of LLMs is also evaluated on the two representative closed-ended text generation tasks: machine translation and text summarization. For machine translation, we follow previous works [22, 70, 69, 30] to evaluate the translation capability on four WMT22 translation tasks (Chinese-to-English, English-to-Chinese, German-to-English, and English-to-German). For text summarization, we follow Liu et al. [30] to conduct the evaluation on CNN/DailyMail Dataset [18]. We employ beam search with a beam size of 4 for machine translation and 2 for summarization. The prompt for machine translation is "Translate the following sentences from [SRC] to [TGT]." The prompt for summarization is "Write a brief and focused summary of the passage that follows.".

## C Effects of $k$ on AR Models

We study the effects of exponent hyper-parameter $k$ on autoregressive models. Table 8 presents the BLEU scores of autoregressive models as the exponent $k$ varies, showing that the optimal performance is achieved when $k = 1$. Other choices of $k$, such as $k = 0.5$ or $0.75$, also yield improvements, predominantly in the context of the greedy search setting.

Table 8: BLEU scores of autoregressive models as the exponent k varies on WMT14 EN-DE test set.

| k-th Power | 0.5 | 0.75 | 1 | 2 | 3 |
|---|---|---|---|---|---|
| Greedy | 26.89 | 26.89 | 26.92 | 26.78 | 26.13 |
| Beam5 | 27.62 | 27.74 | 27.78 | 27.49 | 26.76 |

# D    Correlations on Other NAR Models

In Section 4.2, we present compelling evidence in support of the mode collapse property of convex function effectively mitigating the multimodality issue in the NAR model. This evidence is derived from the strong correlation observed between model entropy and generation fluency in the CMLM model, as demonstrated in Table 4. In this section, we provide additional evidence for other NAR models to further support our findings in Table 9 and 10.

Table 9: Prediction confidence (Output NLL) and generation fluency (External PPL) of Vanilla-NAT on WMT14 EN-DE test set.

| k-th Power | 1 | 2 | 3 | 5 | 8 |
|---|---|---|---|---|---|
| Confidence (Output NLL) $\downarrow$ | 23.34 | 16.17 | 11.32 | 6.25 | 6.27 |
| Fluency (External PPL) $\downarrow$ | 1000.06 | 730.91 | 463.78 | 344.56 | 353.40 |

Table 10: Prediction confidence (Output NLL) and generation fluency (External PPL) of CTC on WMT14 EN-DE test set.

| k-th Power | 1 | 2 | 3 | 5 | 8 |
|---|---|---|---|---|---|
| Confidence (Output NLL) $\downarrow$ | 18.74 | 13.88 | 11.20 | 7.69 | 5.55 |
| Fluency (External PPL) $\downarrow$ | 174.79 | 142.80 | 134.28 | 137.07 | 154.60 |

# E    Results on Alternative Choice of Convex Function

In addition to the exponential function, we have also explored another choice of convex function in our framework of convex-composition loss. In this section, we discuss the results of the choice of power function, i.e., $\mathcal{L}_f(\theta) = -\mathbb{E}_{x \sim p_{data}(x)}[-(-\frac{\log(p_\theta(x))}{T})^k], 0 \leq k \leq 1$. The results obtained from applying the power function in convex-composition loss are presented in Table 11.

Table 11: Results of BLEU scores by applying power function in convex-composition loss. We denote the MLE baseline by using $k = 1.0$. We employ greedy decoding for Transformer. In cases where training fails, we use "N/A" to denote such instances.

| k-th Power | 0.1 | 0.3 | 0.5 | 0.7 | 1.0 |
|---|---|---|---|---|---|
| Transformer | 26.64 | 26.68 | 26.60 | 26.52 | 26.48 |
| Vanilla-NAT | N/A | N/A | 10.74 | 10.51 | 10.41 |

We have observed that the benefits of applying the power function within the convex composition framework are significantly marginal compared to the exponential function, especially in the case of Vanilla-NAT. In addition, we have found the training process may encounter difficulties or failure when $k$ is approaching 0. We attribute such problem to the shape of $f'(g(p_\theta(x)))$ when power function is applied, i.e., $k \cdot (-\frac{\log(p_\theta(x))}{T})^{k-1}$.

As shown in Figure 3, the value of $f'(g(p_\theta(x)))$ will approach a constant 1 as $k$ approaches 1. This phenomenon arises due to the reduction in the convexity of function $f$, resulting in a decrease in gain. In case of $k$ approaching 0, the situation is even worse where $f'(g(p_\theta(x)))$ will experience a sudden increase from an extremely small value near 0. These factors result in an unstable training process and contribute to the power function being less suitable within the framework of convex-composition loss.

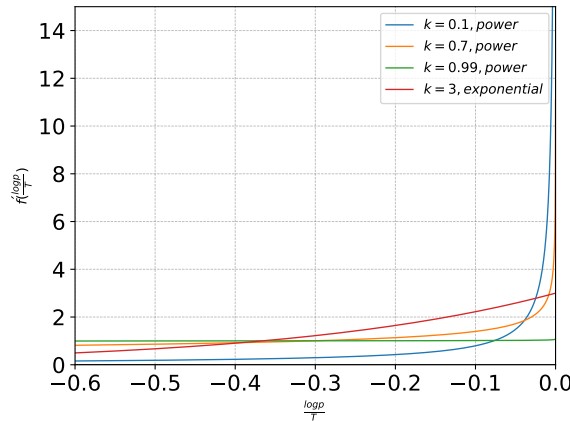

Figure 3: Shapes of $f'(g(p_\theta(x)))$ when different convex functions are applied.

## F Results on Diverse Generation

We study the effects of convex functions on VAE-based text generation models by replacing the log-probability-based reconstruction loss in ELBO with the convex-composition loss. Formally, we train the model using the following loss:

$$\mathbb{E}_{z\sim q(z|x)} - f(\frac{1}{T}\log p(x|z)) + \mathrm{KL}(q(z|x)||p(z)), \tag{29}$$

where we opt for the convex function $f$ to be $e^{kx}, k \geq 0$. We perform experiments within the context of conditional generation, utilizing a VAE-based non-autoregressive model [54, 16] for the task of machine translation. During inference, we randomly sample the latent variable 3 times to generate diverse texts. We assess the quality with BLEU score computed against reference (reference-BLEU) and measure the diversity with BLEU score computed against each other (pairwise-BLEU). The average value and standard derivation are reported in Table 12.

Table 12: Reference-BLEU and Pairwise-BLEU scores of VAE-based NAT models trained with different objectives on WMT14 EN-DE test set. The texts are generated by sampling the latent distribution 3 times.

|  | ELBO | Convex + KL |
|---|---|---|
| **Reference-BLEU** | 16.23±.14 | 23.35±.04 |
| **Pairwise-BLEU** | 29.52±.20 | 91.91±.03 |

During the training process, we have observed that KL divergence tends to vanish more readily when the convex functions are applied. We attribute this phenomenon to the smaller norms of gradients associated with the convex-composition loss. As a result, the gradient of the KL divergence dominates the model update, leading to the KL divergence vanishing. We note VAE-based text generation models trained using the convex-composition loss exhibit a higher generation quality while suffering from poor diversity, which is consistent with the mode collapse property of convex function.

## G Analysis of Convex Learning and Knowledge Distillation

With the ability to capture a concentrated distribution from datasets exhibiting a multi-modal distribution, the proposed convex learning approach shows similar dynamics to knowledge distillation [19], a technique which encourages the student model to imitate the output of the teacher model. To compare the two methods, we utilize autoregressive Transformer as the teacher and apply sequence-level knowledge distillation [23] to construct a dataset of lower complexity, and train the models using different losses.

Table 13: BLEU scores of autoregressive and vanilla-NAT models trained with or without knowledge distillation (KD) on WMT14 EN-DE test set.

|  | Transformer | | Vanilla-NAT | |
|  | MLE | Convex | MLE | Convex |
| --- | --- | --- | --- | --- |
| **w/ KD** | 27.73 | 27.80 | 19.18 | 23.17 |
| **w/o KD** | 27.57 | 27.78 | 10.41 | 16.74 |

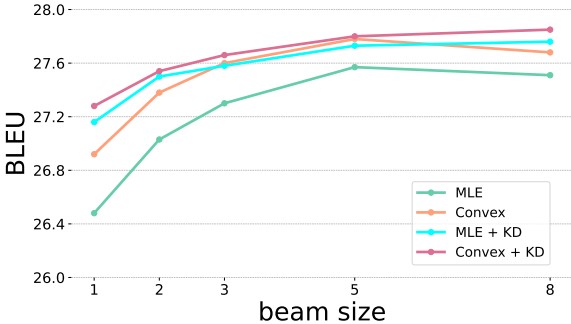

Figure 4: BLEU scores of autoregressive models as beam size varies with or without knowledge distillation (KD) on WMT14 EN-DE test set.

The results in Table 13 and Figure 4 demonstrate that convex learning and knowledge distillation have similar effects on text generation models. Both methods lead to significant improvements on non-autoregressive models and bridge the performance gap between greedy and beam search of autoregressive models. It is worth noting that training with the convex-composition loss avoids the intricate process of training an additional teacher model and decoding the whole training set to achieve the improvements. Moreover, convex-composition loss can be combined with knowledge distillation to further enhance the performance.

