# OpenReview forum: "Beyond MLE: Convex Learning for Text Generation"
_NeurIPS.cc/2023/Conference — NeurIPS 2023 poster_

### Official Review · Reviewer_CKPv · 2023-07-04

**Soundness:** 2 fair
**Presentation:** 3 good
**Contribution:** 2 fair
**Rating:** 5
**Confidence:** 5

**Summary:**

Based on the ideas of convex optimization, the paper proposes a new loss function for the training of text generation models. The experimental results show that it has certain advantages over MLE, whether in autoregressive models or non-autoregressive models.

**Strengths:**

1. The paper has done a relatively detailed theoretical derivation;

2. The final obtained loss is simple and practical;

3. Experimental results show its superiority over MLE in both autoregressive and non-autoregressive models;

4. It has narrowed the gap between greedy search and beam search.

**Weaknesses:**

1. Some key experimental hyperparameters were not provided, such as the value of T, which I could not find in the main text or the appendix;

2. The experimental part only compared with MLE, without comparing other loss functions (such as other classification losses used to approximate accuracy);

3. The notation is a bit confusing (for example, k is used in both formula 11 and theorem 3, but the meanings are not the same);

4. Since the goal of the Convex Loss is one-hot rather than learning distribution, it seems that it can only be used to improve the greedy/beam search, and is not applicable to the decoding strategy of random sampling.


**Questions:**

The original goal of Convex Loss is to learn one-hot, which is more consistent with the greedy/beam search decoding strategy. However, this contradicts gradient-based optimizers, so the final loss balances the goal of one-hot and gradient, which is formula 12. My questions are:

1. Compared to the gradient of MLE, the gradient of formula 12, which is formula 13, seems to be smoother. This appears to make the model's output smoother rather than closer to one-hot. Doesn't this contradict the starting point of Convex Loss?

2. Is it possible to propose a quantitative metric to measure the impact of these two aspects? Based on this metric, we might be able to directly obtain an optimal solution, instead of introducing an adjustable hyperparameter like in formula 12. Because as long as an adjustable hyperparameter is introduced, the experimental results are likely to improve to a certain extent, which is a very commonplace feature and seems to waste the previous detailed theoretical analysis.

**Limitations:**

I couldn't find any discussion about limitations in the main text and the appendix of the paper. Perhaps the authors should include it in the paper.

---

> ### Author Rebuttal · Authors · 2023-08-10
>
> Thank you for your constructive comments. We provide discussions and explanations about your concerns as follows.
>
> > Some key experimental hyperparameters were not provided, such as the value of T, which I could not find in the main text or the appendix; The notation is a bit confusing (for example, k is used in both formula 11 and theorem 3, but the meanings are not the same);
>
> We apologize for the confusion caused by our presentation. T represents the length of target sentence. In the revised version of the paper, we will ensure that all key experimental hyperparameters are adequately presented and explained. To avoid any ambiguity, we will change the notation of "k" to "m" in Theorem 3 and Theorem 4.
>
> We appreciate your attention to detail and valuable feedback, which helps us improve the clarity of our paper.
>
> > Compared to the gradient of MLE, the gradient of formula 12, which is formula 13, seems to be smoother. This appears to make the model's output smoother rather than closer to one-hot. Doesn't this contradict the starting point of Convex Loss?
>
> Thank you for your question regarding the gradient of our loss. We understand your concern and would like to clarify the misunderstanding.
>
> The gradient of formula 12, $fg(p(x))$, is $f'g(p(x)) \cdot g'(p(x))$ (line 242), where formula 13 represents only a part of the gradient. Compared to the gradient of MLE, $g'(p(x))$, our gradient has an additional term, $f'g(p(x))$ (formula 13), which can be understood as the weight for the MLE loss. With $f$ being a convex function, $f'$ is an increasing function, so $f'g(p(x))$ will be larger for samples with greater likelihood. This encourages the model to assign higher probability to the most probable outputs, which aligns with the starting point of Convex Loss.
>
> We hope this clarification resolves your concern. If you have any further questions, please feel free to bring them up for discussion.
>
> > Since the goal of the Convex Loss is one-hot rather than learning distribution, it seems that it can only be used to improve the greedy/beam search, and is not applicable to the decoding strategy of random sampling.
>
> Thank you for your comment regarding the applicability of our approach. You are correct that our method primarily focuses on closed-ended generation tasks, such as machine translation and text summarization, where convex loss can help greedy/beam decoding find high-probability outputs. While our approach ensures a sharper optimal distribution, it currently cannot accurately control the shape of the distribution, which limits its applicability to open-ended generation tasks.
>
> Nevertheless, we believe that further research in this direction could lead to more precise control over the shape of the optimal distribution. This could potentially improve commonly used sampling methods (e.g., top-k sampling, nucleus sampling, temperature sampling) in open-ended generation by enabling more controllable and effective generation of desired results.
>
> > Is it possible to propose a quantitative metric to measure the impact of these two aspects? Based on this metric, we might be able to directly obtain an optimal solution, instead of introducing an adjustable hyperparameter like in formula 12. Because as long as an adjustable hyperparameter is introduced, the experimental results are likely to improve to a certain extent, which is a very commonplace feature and seems to waste the previous detailed theoretical analysis.
>
> Thank you for your very insightful question regarding the possibility of proposing a quantitative metric to measure the impact of the two aspects you mentioned, which are indeed our major principles of loss designing. Your question highlights the subtle trade-off between guiding the model to learn the desirable one-hot distribution and ensuring that the loss is not difficult to minimize.
>
> Finding a quantitative metric to balance these two aspects and obtain an optimal solution would indeed be a very valuable contribution to the field. However, due to the complexity and challenges associated with this task, we are unable to provide a solution within a short period of time. We consider this an important direction for future work and will explore this idea in our ongoing research.

---

### Official Review · Reviewer_hk8n · 2023-07-04

**Soundness:** 3 good
**Presentation:** 3 good
**Contribution:** 3 good
**Rating:** 7
**Confidence:** 4

**Summary:**

Authors have proposed the novel way set of loss functions to learn neural sequence model in both AR and NAR factorization. The core idea is to sharpen the model distribution so that is comes closer to the one-hot target distributions which are typically used for closed-form generation task such as NMT. They have theoretically confirmed how the usage of composition of convex and concave functions with model's probability will result in the aforementioned sharper model's induced distribution.
They performed set of experiments in both AR and NAR setting. AR settings shown that the proposed approach narrows down the gap between greedy and beam search for NMT and slightly improve scores on summarization task. NAR results shows improvements in translation quality.

**Strengths:**

* The proposed apprach is very intuitive and makes sense from the optimization point of view and given the motivation of getting a sharper distribution. It is also easy to implement.

* The proposed framework of convex/concave composition allows community to extend this by examining other potential candidates for the loss function.

* Assuming that the NAR NMT baseline results are strong baselines, the proposed approach substantially improves the NAR prediction quality.

**Weaknesses:**

# Experimental analysis

* The resulting BLEU scores for AR setting are too close and not convincing. I think it is required to perform multiple random seed training or other way of estimating the translation performance variance to verify the improvement. In its current form I would be more convinced that AR modeling performance is nearly the same in terms of translation quality.

# Related work

There is a recent work (I am not an author!) around modeling sharper distributions by reformulating the sequential random variables to be one-hot: https://arxiv.org/abs/2205.00704. I believe it is very similar in terms of the underlying motivation and observed improvements. Given that this was not mentioned in this work, I am not sure how well other related work is covered in this paper.

**Questions:**

# tuning of k in AR vs NAR

* you said you chose k = 1 for AR and tuned K for NAR experiments. And you only shown the ablation with k for NAR task. Why is so? I suspect there are some interesting pieces about k in AR setting. Could you share any insights? Also I believe it is crucial to tell that you tuned k for NAR experiments as it decreases the practicality of this method in real life.

* from training details I see that you didnt use label smoothing for NAR modeling with convex objective. Why is so? I think that the answer "it works better empirically" is not good enough here as this method inherently related to making sharper distributions while smoothing is doing the opposite. So I tend to believe there is some interesting relationship there.

**Limitations:**

Authors did not discuss limitations in the main text of their submission.

---

> ### Author Rebuttal · Authors · 2023-08-10
>
> Thank you for your constructive comments. We provide discussions and explanations about your concerns as follows.
>
> > The resulting BLEU scores for AR setting are too close and not convincing. I think it is required to perform multiple random seed training or other way of estimating the translation performance variance to verify the improvement. In its current form I would be more convinced that AR modeling performance is nearly the same in terms of translation quality.
>
> Thank you for your feedback regarding the evaluation on AR setting. Following your suggestion, we plan to perform multiple random seed training and report the results in the revised version of the paper. We would also like to emphasize that the current version of our paper has already demonstrated the positive effect of our method on autoregressive models:
>
> 1. We conducted AR experiments on both machine translation and text summarization tasks. While the performance is close in the machine translation with beam search setting, our method shows improvements in other settings.
>
> 2. Figure 2 demonstrates a consistent improvement of our method across different beam sizes. The gap between our method and the baseline narrows as the beam size increases, which complies with our theoretical analysis and suggests that the convex loss has a positive impact on translation quality.
>
> 3. In the attached response PDF, we provide the AR performance under different values of k. We observe consistent improvements when k is close to 1, especially for greedy search.
>
> We hope this response addresses your concern and demonstrates the effectiveness of our proposed method. If you have any further questions or concerns, please feel free to bring them up for discussion.
>
> > There is a recent work (I am not an author!) around modeling sharper distributions by reformulating the sequential random variables to be one-hot: https://arxiv.org/abs/2205.00704. I believe it is very similar in terms of the underlying motivation and observed improvements. Given that this was not mentioned in this work, I am not sure how well other related work is covered in this paper.
>
> We appreciate your reference to the recent work (https://arxiv.org/abs/2205.00704) that shares a similar motivation with our paper in terms of modeling sharper distributions. We apologize for not mentioning this work in our paper and would like to express our gratitude for bringing it to our attention.
>
> Upon careful analysis of the method used in this paper, we found that it indeed shares a similar motivation with our work. However, the proposed method in the mentioned paper differs significantly from ours. The method in the paper reformulates the distribution of the model at the word level, which may not directly facilitate the model in finding high probability sentences at the sentence level. In contrast, our method directly trains the model to focus on highly probable sentences.
>
> Nonetheless, we believe that the mentioned paper is an interesting and relevant work. In the revised version of our paper, we will include a discussion of this related work and highlight the differences between our method and the one proposed in the mentioned paper. Thank you once again for your valuable feedback.
>
> > you said you chose k = 1 for AR and tuned K for NAR experiments. And you only shown the ablation with k for NAR task. Why is so? I suspect there are some interesting pieces about k in AR setting. Could you share any insights? Also I believe it is crucial to tell that you tuned k for NAR experiments as it decreases the practicality of this method in real life.
>
> Thank you for your question regarding the choice of k. The reason we did not show the ablation study with k for the AR setting is actually the space constraints. In the attached response PDF, we supplement the results for different values of k in the AR setting. Our results show that k=1 performs the best, while other choices of k, such as k=0.5 or 0.75, also bring improvements, especially in the greedy search setting.
>
> For the NAR experiments, we tuned k from the set {1, 2, 3, 5, 8} on the validation set. We also provide the test set results for different values of k in Figure 2. In the revised version of the paper, we will include the results of different k in the AR setting to ensure a clearer presentation.
>
> > from training details I see that you didnt use label smoothing for NAR modeling with convex objective. Why is so? I think that the answer "it works better empirically" is not good enough here as this method inherently related to making sharper distributions while smoothing is doing the opposite. So I tend to believe there is some interesting relationship there.
>
> Thank you for your insightful question regarding the relationship between label smoothing and convex objective. We have provided the experimental results in our attached response PDF, showing that autoregressive models perform better with label smoothing, while non-autoregressive models work better without it. As you mentioned, this experimental phenomenon is indeed interesting. Our interpretation is as follows:
>
> Label smoothing serves as a regularization technique, helping the model generalize better on test data. Although we aim for a sharp distribution in autoregressive models to facilitate approximate search algorithms in identifying the most likely output, the importance of regularization may outweigh the need for a sharp distribution, making label smoothing beneficial.
>
> However, for non-autoregressive models, they are theoretically unable to fit the data distribution, let alone overfit the dataset. In this case, finding a sharp distribution for NAR models to fit becomes more important than preventing overfitting with label smoothing. As a result, NAR models prefer not to use label smoothing.

---

> > ### Comment · Reviewer_hk8n · 2023-08-18
> > **thank you**
> >
> > thanks for your effort in doing extra experiments and providing more details to address my concerns. My concerns are addressed now and i am going to increase my score!

---

### Official Review · Reviewer_pbsn · 2023-07-04

**Soundness:** 4 excellent
**Presentation:** 4 excellent
**Contribution:** 4 excellent
**Rating:** 7
**Confidence:** 3

**Summary:**

The objective of the authors in this paper is to develop a machine translation model that doesn't need to learn the complete output distribution given the input, but instead focuses on highly probable outputs based on p_data. To achieve this, the authors suggest replacing the logarithm in Maximum Likelihood Estimation (MLE) with a convex increasing function. They demonstrate that optimizing such a function leads to a desirable one-hot distribution. However, the authors acknowledge that training such a model can be challenging due to the gradient being proportional to the probability of the samples, which can be quite low for sequences.

To address this issue, the authors propose combining a concave function (e.g., log in MLE) with an increasing convex function. They illustrate that optimizing this composite function yields a distribution that assigns higher probability to outputs with greater likelihood under p_data.

The authors apply their loss function to train both autoregressive and non-autoregressive machine translation models. They demonstrate that the resulting greedy and beam outputs from these models surpass the performance of the greedy and beam outputs produced by MLE.

Update: I have read the author's response. I am satisfied with the response and since I had already given an accept to the paper, I will keep my ranking.

**Strengths:**

The authors' approach to addressing the challenge of learning a peaky distribution is innovative. The novel convex-composition loss functions introduced in this paper are particularly interesting within the realm of sequence generation.

The paper is highly compelling and offers an intuitive reading experience. The authors' strategy of incorporating an increasing convex function to emphasize high-probability outputs under p_data is remarkable.

The theoretical concepts discussed in this paper hold immense potential impact, even if the direct application to machine translation does not yield substantial results.

**Weaknesses:**

Equation (5) may not hold true for non-autoregressive models as these models employ latent variables that interconnect the output tokens, as mentioned in [1] [2].

The term "Convex loss functions" in section 3.2 is misleading since the negative logarithm used in MLE is, in fact, a convex loss function. The authors actually refer to "Loss functions with convex f."

There is an inaccuracy in line 200 where the authors state that the standard loss function in maximum likelihood estimation is the log-probability. This is incorrect. The loss function is actually the negative log-probability, which is convex and not concave.

Equation (12) can only utilize odd values for the power function, as for even values, the function decreases within the range of g(p_theta(x)).

[1] Gu, J., et al. "Non-autoregressive neural machine translation." International Conference on Learning Representations (ICLR). 2018.
[2] Gu, Jiatao, and Xiang Kong. "Fully Non-autoregressive Neural Machine Translation: Tricks of the Trade." Findings of the Association for Computational Linguistics: ACL-IJCNLP 2021. 2021.

**Questions:**

In line 149, the statement "If Lf has multiple optimal distributions, we simply arrange them in alphabetical order and take the first one to be pf" raises concerns. It is not feasible to arrange distributions in alphabetical order since the space of distributions is not countable, even if the sample space is countable.

Regarding step number 3 in equation (18) of the Appendix, it is unclear how the optimality of pg implies its validity. Further clarification is needed to establish the reasoning behind this step and its connection to the optimality of pg.

**Limitations:**

The authors have discussed that convex functions of probability are hard to train. It will be good to have a similar discussion about the composition loss functions discussed in the paper.

---

> ### Author Rebuttal · Authors · 2023-08-10
>
> Thank you for your constructive comments. We provide discussions and explanations about your concerns as follows.
>
> > Equation (5) may not hold true for non-autoregressive models as these models employ latent variables that interconnect the output tokens.
>
> Thank you for your comment regarding Equation (5) and its applicability. We acknowledge that the equation may not hold true for latent models, as they employ latent variables that interconnect the output tokens. In section 2.2, we focus on introducing the background knowledge about text generation models, which is why we used the original form in Equation (5). We appreciate your feedback and will clarify this point in the revised version of the paper to avoid any confusion.
>
> > The term "Convex loss functions" in section 3.2 is misleading since the negative logarithm used in MLE is, in fact, a convex loss function. The authors actually refer to "Loss functions with convex f". There is an inaccuracy in line 200 where the authors state that the standard loss function in maximum likelihood estimation is the log-probability. This is incorrect. The loss function is actually the negative log-probability, which is convex and not concave.
>
> Thank you for pointing out this issue. In the revised version of the paper, we will avoid using the term "Convex loss functions" and instead refer to them as "Loss functions with convex f" to prevent any confusion. Additionally, we will correct the statement in line 200 to "The standard loss function in maximum likelihood estimation is the negative log-probability, where log-probability is a concave function."
>
> > Equation (12) can only utilize odd values for the power function, as for even values, the function decreases within the range of g(p_\theta(x)).
>
> Thank you for pointing out the issue with the power function in Equation (12). We acknowledge that there was a writing mistake in our paper, which has been fixed in Appendix D. As you correctly noted, the power function $f(x)=x^k$ is not suitable for our purpose due to its behavior within the range of $\log(p)$. For even values of k, the function decreases, and for odd values of k, it is concave.
>
> The correct form should be $f(x)=-(-x)^k$, with $0 < k < 1$, which is increasing and convex when $x < 0$. The corrected form and its experiment results are presented in Appendix D.
>
> We apologize for the confusion caused by this mistake and appreciate your attention to detail. We hope this clarification resolves your concern.
>
> > In line 149, the statement "If Lf has multiple optimal distributions, we simply arrange them in alphabetical order and take the first one to be pf" raises concerns. It is not feasible to arrange distributions in alphabetical order since the space of distributions is not countable, even if the sample space is countable.
>
> We appreciate your attention to detail and agree that our original statement was not appropriate. We will update the statement as follows: "If Lf has multiple optimal distributions, we use pf to denote an arbitrary optimal distribution. This choice does not harm the generality of our analysis, as the subsequent discussion is applicable to all optimal distributions."
>
> > Regarding step number 3 in equation (18) of the Appendix, it is unclear how the optimality of pg implies its validity. Further clarification is needed to establish the reasoning behind this step and its connection to the optimality of p_g.
>
> Thank you for your valuable question. We apologize for any confusion caused by our presentation, as we skipped too many steps in the derivation of Equation (18).
>
> We can prove Equation (18) by contradiction, which is simialr to the proof of Theorem 2. If Equation (18) does not hold, then we can further reduce the loss, which contradicts the optimality of $p_g$. Specifically, we can let $p_g'(x_i)=p_g(x_i)-\alpha$, and  $p_g'(x_j)=p_g(x_j)+\alpha$. The gradient of loss with respect to $\alpha=0$ is $p_{data}(x_i) \cdot g'(p_{g}(x_i)) - p_{data}(x_j) \cdot g'(p_{g}(x_j)) < 0$, so we can further reduce the loss with a positive $\alpha$, proving Equation (18) by contradiction.
>
> We hope this clarification resolves your concern. We will update the appendix in the revised version of the paper to include the missing steps and ensure a clearer presentation of the proof.

---

### Official Review · Reviewer_MNhM · 2023-07-06

**Soundness:** 2 fair
**Presentation:** 3 good
**Contribution:** 2 fair
**Rating:** 3
**Confidence:** 4

**Summary:**

This paper proposed a new learning objective for language modeling, which is extended from Maximum likelihood estimation (MLE). More specifically, the author replaced the log function in log-likelihood with an arbitrary increasing function f. Then the author analyzed the properties of the loss under scenarios where f is convex and its relaxation. Besides, the author provides two practically feasible function forms of f for the learning objectives. Experiments with exponential function show empirical improvement of transformers on translation and summarization tasks.

**Strengths:**

1. The paper proposed a variant of MLE loss for text generation, which is novel.

2. The assumptions for theoretical analysis are reasonable and commonly appeared in real-world application scenarios.

3. The paper is clearly stated and easy to follow.

**Weaknesses:**

1. About Baselines: The method should have been evaluated on more baselines. For AR model, the author only compared the proposed method with the vanilla transformer. For NAR, only two baselines (CMLM, CTC) are compared. The method needs to be applied on more powerful baselines (e.g. [1]) to show its effectiveness and universality.

2. About the f function: The author proposed two practical forms for the f function. However, in the experiments, only the exponential form is discussed.

3. About the theoretical analysis: The analyses seem trivial, and provided little instruction on the practical design of the loss function, or to guarantee the effectiveness.

Reference:
[1]. Bridging the Gap between Training and Inference for Neural Machine Translation, ACL 2019


**Questions:**

1. In equation 10， is x_i a sequence or a token? In my understanding, x_i represents a sequence. If x_i is a sequence, the difference between the proposed method and MLE is adding a multiplier to the gradient, as shown in equation 9?

2. What are the empirical benefits of the theoretical analyses in the paper for further applications?

**Limitations:**

The author didn't mention any societal impact of the proposed method.

---

> ### Author Rebuttal · Authors · 2023-08-10
>
> Thank you for your constructive comments. We provide discussions and explanations about your concerns as follows.
>
> > About Baselines: The method should have been evaluated on more baselines. For AR model, the author only compared the proposed method with the vanilla transformer. For NAR, only two baselines (CMLM, CTC) are compared. The method needs to be applied on more powerful baselines (e.g. [1]) to show its effectiveness and universality.
>
> We appreciate your suggestion to evaluate our method on more baselines. In Table 2 of the attached PDF document, we present the outcomes of implementing the proposed convex loss on data generated via sequence-level distillation. Upon applying the convex loss on the distilled training data, we observed a significant improvement in the NAR model, while the effect seems minor on the AR model. Particularly, NAR models benefit from convex loss by acquiring a more concentrated distribution to address their inherent expressive limitations. The simplification of the data distribution, although beneficial, might not suffice for NAR models to achieve this objective. The introduction of our proposed convex loss serves to further facilitate this desired outcome.
> Moreover, we would like to emphasize that our research is centered around the development of a theory-grounded algorithm to learn a concentrated distribution from a multi-modal data distribution. Therefore, we did not compare our methods with other training-augmented techniques, such as oracle-based schedule sampling in [1].
>
> [1]. Bridging the Gap between Training and Inference for Neural Machine Translation, ACL 2019
>
>
> > About the f function: The author proposed two practical forms for the f function. However, in the experiments, only the exponential form is discussed.
>
> Thank you for your comment regarding the f function. We apologize for any confusion caused by the presentation. In our paper, we indeed explored both forms of the f function. However, due to space constraints, we included the results and discussion for the power form in the supplementary materials.
>
> We found that the power form of the f function encountered some difficulties during training, leading to worse performance compared to the exponential form. In the revised version of the paper, we will make it clearer that the results for both forms of the f function are provided and ensure that the supplementary materials are more easily accessible for readers interested in the details of the power form.
>
> > About the theoretical analysis: The analyses seem trivial, and provided little instruction on the practical design of the loss function, or to guarantee the effectiveness. empirical benefits of the theoretical analyses
>
> Thank you for your feedback regarding the theoretical analysis in our paper. We would like to emphasize that our analysis is non-trivial and contributes novel insights to the field. Our work presents four theorems accompanied by detailed proofs, which were previously unknown in the field.
>
> Our theoretical analysis provides valuable practical guidance for the design of loss functions. It demonstrates the effectiveness of convex functions in learning a one-hot distribution, while also highlighting the limitations of directly optimizing convex functions due to gradient-related challenges. This understanding informed the development of our composite loss function, a key step of our work.
>
> Furthermore, our theoretical analysis guarantees the effectiveness of the composite loss function by showing that it yields a distribution that assigns higher probability to outputs with greater likelihood under p_data. This property is particularly desirable for closed-ended text generation tasks, where generating the most appropriate response is the primary goal.
>
> We hope that this response clarifies the importance and contributions of our theoretical analysis. We appreciate your feedback and will continue refining our paper to make our contribution clear.
>
> > is x_i a sequence or a token?  the difference between the proposed method and MLE is adding a multiplier to the gradient, as shown in equation 9?
>
> Thank you for your question regarding Equation 10. We would like to confirm that x_i indeed represents a sequence, not a token. Your understanding of the difference between the proposed method and MLE is correct. In our method, we add a multiplier to the gradient, which can be interpreted as a weight for the loss.
>
> As discussed in lines 242-247 and shown in Equation 13, the gradient of the convex-composition function is f'g·g'. The additional term f'g serves as a weight for the loss. Given that f is a convex function and g is an increasing function, the weight f'g is larger for more probable samples, thereby directing the model's focus towards generating outputs with high probabilities.

---

### Official Review · Reviewer_v7Us · 2023-07-07

**Soundness:** 4 excellent
**Presentation:** 4 excellent
**Contribution:** 4 excellent
**Rating:** 7
**Confidence:** 5

**Summary:**

This paper discusses the potential limitations of likelihood/KL-guided training for natural language generators from a measure theoretical perspective. It analyses the discrepancy between maximizing the likelihood (thus recall-friendly) and producing high-quality samples (thus precision-oriented) in autoregressive and non-autoregressive models, and proposes an alternative to MLE as the concrete solution. Experiments on enhancing autoregressive/non-autoregressive language models on machine translation validate the effectiveness of the proposed objective.

**Strengths:**

1. The theoretical analysis is insightful, solid, and well-presented. It makes a lot of sense and the corresponding experiment results comply with the analysis, especially the ones on non-autoregressive models.

2. The measure theoretical problem it studies has been a critical yet long-neglected one since the falling short of language GANs. This paper is a solid step towards the ultimate solution and I would really hope the publication of this paper can bring the community's focus back to this area.

**Weaknesses:**

1. It's less analyzed in this paper about the proposed objective on some potential (important and necessary) scenarios. For example, for text VAE/other types of latent-variable text generators, it's long been a dilemma to improve the generation quality while preventing the latent collapse problem.

**Questions:**

1. As stated in Weaknesses, is it possible to include additional experiments on diverse text generation with VAEs? You can try interpolation and I'm very curious if the proposed approach is eventually capable of mitigating the latent collapse of text VAEs.

2. I would appreciate it if you can explicitly compare against models under setups with/without sample-based distillation from teacher models. As far as I'm concerned the dynamics of distillation (especially in machine translation) would be somehow similar to the proposed approach, I'd like to see if it is possible to achieve the performance of a model w/ distillation by simply replacing the objective with the proposed one.

Would be more than happy to raise my scores if these two aspects are addressed in the revision period.

**Limitations:**

I don't observe any potential societal limitations of this work. This work is more of an algorithmic/theoretical contribution to the community.

---

> ### Author Rebuttal · Authors · 2023-08-10
>
> Thank you for your constructive comments. We provide discussions and explanations about your concerns as follows.
>
> > Include additional experiments on diverse text generation with VAEs. You can try interpolation and I'm very curious if the proposed approach is eventually capable of mitigating the latent collapse of text VAEs
>
> Following your advice, we study the effects of convex functions on VAE-text models by replacing the log-probability-based reconstruction loss in the ELBO with the convex loss. Formally, we train the model using the following loss:
>
> $\mathbb{E}_{z\sim q(z|x)} -f(\frac{1}{T}\log p(x|z)) + \mathrm{KL}(q(z|x)||p(z))$.
> Here, we opted for the convex function $f$ to be $\exp(x)$, as outlined in our paper.
>
> We conduct the experiements in a conditional generation scenario, using a VAE-based non-autoregressive model [1,2] applied to a machine translation task. During generation, we randomly sampled the latent variable using 3 different seeds to obtain the texts. We assess the generation quality with the BLEU score computed against reference (reference-BLEU) and measured the diversity with the BLEU score computed against each other (pairwise-BLEU). The average value and standard derivation are reported in the attached response PDF.
>
> During the training process, we observed that the KL divergence tends to vanish more readily when the convex functions are applied. We attribute this phenomenon to the smaller norms of gradients associated with the convex-composition loss. As a result, the gradient of the KL divergence dominates the model update, leading to the vanishing KL divergence. We noted VAE-text models trained using the convex-composition loss exhibit a higher generation quality while suffering from poor diversity, which is consistent with the mode collapse property of convex loss.
>
> [1] Shu, Raphael, et al. "Latent-variable non-autoregressive neural machine translation with deterministic inference using a delta posterior." Proceedings of AAAI 2020.
>
> [2] Gu, Jiatao, and Xiang Kong. "Fully Non-autoregressive Neural Machine Translation: Tricks of the Trade." Findings of ACL-IJCNLP 2021.
>
> >Compare against models under setups with/without sample-based distillation from teacher models. As far as I'm concerned the dynamics of distillation (especially in machine translation) would be somehow similar to the proposed approach, I'd like to see if it is possible to achieve the performance of a model w/ distillation by simply replacing the objective with the proposed one.
>
> Thank you for your insightful comments. We agree with you that the dynamics of distillation is similar to our approach. In sequence-level knowledge distillation, the student model is encouraged to imitate the output of the teacher model, which has a less diverse training dataset and thereby tends to predict a sharper distribution. This characteristic is similar to the effect of convex loss.
>
> To compare the two methods, we used the autoregressive Transformer as the teacher and applied sequence-level knowledge distillation to train our models. The experiment results in the attached response PDF (Table 3, Figure 1) show that convex loss and knowledge distillation have similar effects on text generation models. Both methods lead to significant improvements in non-autoregressive models and bridge the performance gap between greedy and beam search of autoregressive models. It is worth noting that our method does not require training an additional teacher model and decoding the training set to achieve these improvements. Moreover, convex loss can be combined with knowledge distillation to further enhance the performance.
>
> If you have any further thoughts, please feel free to bring them up, and we will be more than happy to engage in an in-depth discussion.

---

> > ### Comment · Reviewer_v7Us · 2023-08-12
> > **Response**
> >
> > I appreciate the efforts from the authors. I'm satisfied about the second-round feedbacks from the authors and I'm ready to raise my score.

---

### Author Rebuttal · Authors · 2023-08-10

We thank reviewers for their valuable feedback and insightful comments on our paper. In response to the concerns and recommendations, we have conducted additional experiments and provided supplementary results in the attached response PDF. In Table 1, we present the results of tuning the hyper-parameter k for autoregressive models. In Table 2, we show the results of models trained with and without knowledge distillation. In Table 3, we report the outcomes of combining our method with text VAEs. In Table 4, we provide the results of models trained with and without label smoothing. Additionally, in Figure 1, we illustrate the performance of autoregressive models trained with and without knowledge distillation across different beam sizes.

---

### Author Response · Authors · 2023-08-21

Dear Reviewers,

We would like to express our gratitude for your time and efforts in reviewing our paper. As a reminder, we will be unable to respond after the author-reviewer discussion period ends on August 21st, 1pm EDT. Please feel free to reach out if you have any follow-up questions or concerns regarding our rebuttal.

Thank you once again for your valuable feedback.

Best regards,
Authors

---

> ### Comment · Area_Chair_JqrX · 2023-08-21
> **Thank you for your rebuttals**
>
> Dear authors,
>
> Thank you for your hard work on the responses. I found especially your clarification of some of the notation and terminology valuable. Even where the reviewers have not responded yet, I assure you that what you wrote will be taken into account in our discussions and decisions.
>
> Best
>
> the ac

---

### Decision · Program_Chairs · 2023-09-21

**Decision:**

Accept (poster)

**Comment:**

The paper studies some constructions for loss functions that mitigate some of the downsides of standard MLE losses in text generation. Through careful function composition, useful loss functions are obtained.

Reviewers find the construction novel and interesting both in theory and in practice. The experimental section has some limitations that the authors have improved with some additional experiments, including one on VAEs; overall I believe the evaluation is relatively convincing. The writing is mostly clear, but I strongly urge the authors to clarify the ambiguity of convexity/concavity wrt common use in the context of classification losses, as pointed out by the reviewers. (The MLE / "cross-entropy" loss is convex w.r.t. the scores).

Overall, this work is of interest to the community and I recommend acceptance. Please take into account all feedback.